# From Time-Series to Text: Multimodal and Agentic Approaches for Automated Drilling Report Generation

**Ankush Mishra** [* 1]  **Sebastiaan Buiting** [* 1]  **Soumyadipta Sengupta** [1]  **Abdallah Benzine** [* 1]

## Abstract

Daily Drilling Reports (DDRs) summarize complex operations by analysing high-frequency sensor data. Automating DDR generation is challenging, requiring joint reasoning over multivariate time series and domain-specific semantics.

In this work, we evaluate three methodologies for DDR generation from raw sensor data: (i) a multimodal architecture integrating time series foundation models (TSFMs, e.g., Chronos-2) via cross-modal conditioning, (ii) a vision-language model (VLM) reasoning over visualized sensor plots, and (iii) a tool-augmented Agentic approach utilizing SQL for iterative data exploration.

Our results reveal complementary strengths. The Agentic system achieves the highest accuracy in precise numerical extraction due to explicit tool use. In contrast, the TSFM-based multimodal approach excels at capturing temporally extended contextual events, demonstrating stronger implicit temporal understanding. Rather than identifying a single best method, these findings establish DDR generation as a multi-paradigm problem and point to hybrid systems—combining agentic retrieval with TSFM inductive biases—as a promising direction for future work.

## 1. Introduction

Industrial operations generate vast volumes of multivariate sensor data that must be distilled into structured reports. In drilling, Daily Drilling Reports (DDRs) condense 24 hours of telemetry into a few telegraphic lines logging the operations, equipment, anomalies, and measurements that define the wellbore's state. These safety-critical reports are still authored largely by hand. Automating their generation is a demanding data-to-text task ([18]; [16]; [17]; [14]), requiring joint reasoning over long, high-frequency time series, discrete operational states, and domain-specific semantics.

The problem imposes two competing demands. A DDR must be numerically exact—depths, flow rates, and pressures are reported to operational tolerance, and a single fabricated value misrepresents the well. Yet it must also be contextually faithful: naming the operation and equipment requires recognizing temporally extended signatures—a casing run, a cementing job, a stuck-pipe event—from raw signals alone, with no activity labels at inference time. Precise extraction and holistic interpretation are distinct competencies that no single paradigm obviously satisfies.

Prior work addresses pieces of this in isolation. *LiveDrill* ([20]) frames DDR generation as a segment-grounded multimodal task but remains limited on rare operations and numerical grounding. Time-Series Foundation Models (TSFMs) ([3]; [2]; [8]; [22]; [9]) learn strong temporal representations but do not target structured reporting, and recent work embedding TSFM encoders in multimodal LLMs ([12]) targets general reasoning, not the rigid constraints of industrial reporting.

These suggest three complementary paradigms. Multimodal architectures ([1]; [13]; [6]) offer implicit temporal priors over the raw signal; VLMs capture procedural structure but struggle with numerical precision ([15]; [4]); and tool-augmented agents ([23]; [19]) query data explicitly for verifiable correctness at the cost of speed. Yet they have never been compared head-to-head on a real-world reporting task.

We close this gap by comparing a multimodal architecture embedding TSFMs (e.g., Chronos-2), a plot-based VLM, and a tool-augmented agent performing iterative SQL exploration. Evaluated on 22,483 industrial operations against expert-authored reports, they reveal a clean functional division of labor: no single paradigm dominates. The agent is unmatched at numerical extraction; the multimodal TSFM more reliably captures temporally extended operational states and equipment metadata at two orders of magnitude lower cost; and the VLM forms a lower bound set by lossy plot encoding. DDR generation is thus inherently multi-paradigm, highlighting the motivation for hybrid systems.

---

[*]Equal contribution  [1]AIQ, Abu Dhabi. Correspondence to: Ankush Mishra <ankush.mishra@aiq.ae>, Abdallah Benzine <abdallah.benzine@aiq.ae>.

*Proceedings of the 43rd International Conference on Machine Learning*, Seoul, South Korea. PMLR 306, 2026. Copyright 2026 by the author(s).

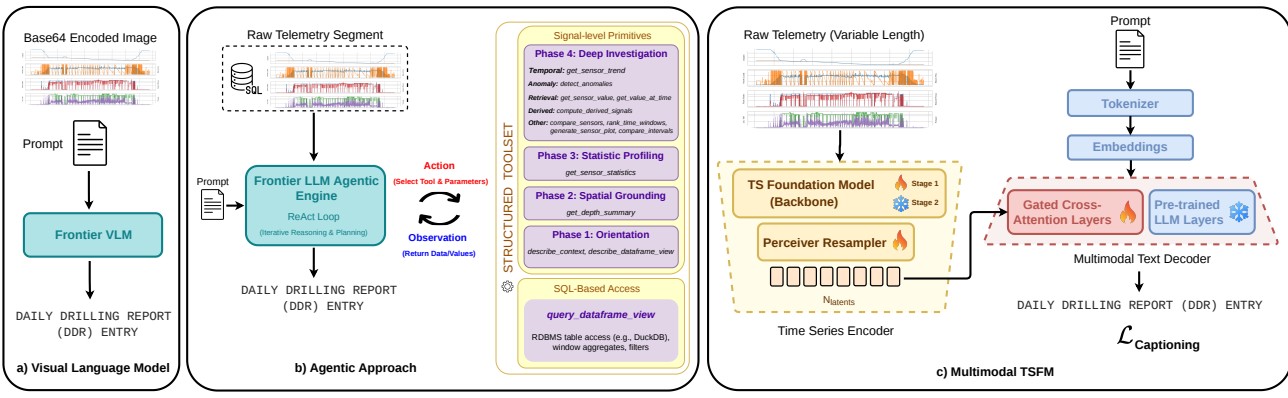

*Figure 1.* Architectural overview of the three evaluated approaches for DDR generation: (a) a **VLM** processing base64-encoded sensor plots; (b) an **Agentic approach** utilizing tool-augmented reasoning to query raw telemetry via SQL and Python; and (c) the **Multimodal TSFM** architecture, which integrates frozen Time-Series Foundation Models (e.g., Chronos-2) with a text decoder via gated cross-attention.

## 2. Problem Formulation

We model DDR generation as a sensors-to-text problem.

**Input.** Let $\mathbf{X} \in \mathbb{R}^{T \times C}$ denote multivariate sensor data with $T$ timesteps and $C$ channels. We are given a set of pre-segmented intervals $\mathcal{A} = \{a_i\}_{i=1}^N$, where each segment $a_i$ defines a unique time window over $\mathbf{X}$. Each segment is associated with exactly one report entry.

**Output.** For each segment $a_i$, the goal is to generate a textual entry $y_i^{text}$ describing the operation, including both contextual interpretation and relevant numerical values. We define the complete report as the sequence $\mathbf{Y} = \{y_i^{text}\}_{i=1}^N$.

**Objective.** We learn a mapping $f_\theta : (\mathbf{X}, \mathcal{A}) \rightarrow \mathbf{Y}$ that produces accurate and concise reports.

## 3. Approaches

### 3.1. Multimodal TSFM-Based Architecture

We adapt the *OpenTSLM-Flamingo* (12) architecture to bridge continuous time-series telemetry and discrete textual domain. As illustrated in Figure 1c, we replace the original patch-based CNN with high-capacity TSFMs, such as **Chronos-2** and **Toto** (7), to serve as the foundational encoder backbone.

The TSFM extracts temporal latent vectors, which are subsequently compressed by a Perceiver Resampler into fixed-length tokens. These tokens are integrated into a Gemma3-1B text decoder using Gated Cross-Attention (GCA) layers (1). Crucially, to preserve the foundational reasoning capabilities of the language model, all default LLM decoder layers are kept strictly frozen throughout the entire training process; only the newly interleaved GCA layers are trainable. This configuration enables the LLM to dynamically attend to multivariate sensor data while preserving the integrity of its textual context window. Refer to Appendix C

for further architectural and training specifics.

**Multi-stage Training Strategy.** To optimize this architecture for industrial reporting without catastrophic forgetting, we employ a two-stage training strategy.

*Stage 1: Operational Pre-training.* We establish operational context through parallel classification tasks. The full architecture undergoes *global alignment* to condition the cross-attention layers to sensor inputs, while the TSFM backbone undergoes independent *refinement* for high-fidelity feature extraction.

*Stage 2: Label-Grounded Captioning.* For final DDR generation, the overall model is first initialized using the global alignment weights from Stage 1. We then specifically load the TSFM backbone with its independently refined weights and keep it strictly frozen throughout this second training phase. To anchor caption generation in the operational context, we enforce a classification-grounded output format: ⟨*Class label*⟩ *DDR:* ⟨*DDR*⟩

### 3.2. Vision-Language Approach

While foundational time-series models process numerical sequences directly, VLMs have demonstrated remarkable efficacy in chart reasoning (21). To evaluate this, we reframe DDR generation as a visual question-answering task, utilizing GPT-5 as the core reasoning engine (Figure 1a).

In this setup, the multivariate sensor data $\mathbf{X}_i$ for a given segment $a_i$ is rendered into a 2D visual plot. These plots are fed to the VLM alongside a structured system prompt containing anti-hallucination constraints and stylistic reference examples. This transforms the time-series interpretation task into a visual chart understanding problem.

To determine the optimal grounding strategy, we also evaluated various prompting configurations using matched image-text pairs; a comprehensive analysis of these ablation studies

*Table 1.* Comparative Model Performance Across Rubric Criteria and Operational Classes, Alongside Generation Costs.

| Model | Eval (%) | Cost / Gen ($) | Evaluation Score (%) Across Rubrics | | | | | | | | Evaluation Score (%) Across Operational Classes | | | | | | | | |
|---|---|---|---|---|---|---|---|---|---|---|---|---|---|---|---|---|---|---|---|
| | | | A | B | C | D | E | F | G | H | CM | CMT | CORE | CSG | DRILL | DRLOUT | REAM | STKP | TRIP |
| | | | *Rubric Weight (%)* | | | | | | | | *Sample Distribution (%)* | | | | | | | | |
| | – | – | *30* | *15* | *2* | *15* | *15* | *5* | *3* | *15* | *2.4* | *7.8* | *0.6* | *12.7* | *32.1* | *4.5* | *4.4* | *1.3* | *34.3* |
| GPT-5 (VLM) | 55.4 | 0.02677 | 62 | 65 | **100** | 37 | 84 | 32 | 49 | 25 | 47.0 | 54.4 | 55.8 | 41.8 | 61.7 | 50.6 | 52.1 | **53.4** | 54.4 |
| GPT-5 Agentic | **70.1** | 0.14458 | **71** | **97** | **100** | **51** | **87** | 37 | 54 | 54 | 51.9 | 57.9 | **70.2** | 50.6 | **85.5** | **66.0** | **72.5** | 50.7 | **67.5** |
| MM-TSFM (CNN) | 63.7 | **0.00112** | 62 | 69 | 99 | 44 | 86 | **53** | 67 | **58** | **52.6** | 63.2 | 53.5 | 52.3 | 78.4 | 57.8 | 48.9 | 41.3 | 59.0 |
| MM-TSFM (Chronos2) | 67.5 | 0.00126 | 68 | 78 | 98 | 47 | 85 | **53** | **71** | **58** | 48.0 | **65.1** | 57.4 | **58.9** | 82.9 | 50.5 | 56.1 | 41.7 | 62.7 |
| MM-TSFM (Toto) | 63.7 | 0.00129 | 64 | 72 | 98 | 45 | 86 | 46 | 69 | 53 | 48.8 | 57.1 | 43.3 | 51.4 | 78.6 | 55.1 | 50.4 | 42.1 | 60.8 |

and prompt templates is provided in Appendix D.

### 3.3. Agentic Approach

We propose a tool-augmented Agentic approach (Figure 1b) in which GPT-5 generates DDR entries by iteratively querying raw sensor data through a structured toolset. This separates *reasoning* from *retrieval*: the model plans which signals to inspect, delegates data access to dedicated tools, and refines its interpretation before committing to a final DDR string.

Given a segment, the agent executes a ReAct-style (23) tool-calling loop. The toolset provides two classes of access: SQL-based access via DuckDB for full window aggregates and filters, and signal-level primitives for higher-level operations like anomaly detection and piecewise trend fitting. To ensure numerical fidelity, the system prompt enforces a strict anti-hallucination constraint, requiring that all reported values be explicitly verified by at least one tool output. Extended implementation details, including the full toolset definitions and agentic prompting templates, are provided in Appendix E.

## 4. Experimental Setup

### 4.1. Data

We evaluate all approaches on a large-scale industrial dataset comprising 22,483 distinct operational events collected from multiple wells. Each sample consists of a temporal window of multivariate sensor data, $\mathbf{X}_i \in \mathbb{R}^{T_i \times C}$, featuring $C = 11$ surface measurement channels (e.g., hook load, standpipe pressure, and rotary speed) sampled at 1 Hz. Because operational durations vary widely, the sequence length $T_i$ fluctuates significantly, with an average length of roughly 11,830 timesteps and a maximum of 86,401 timesteps.

Crucially, each method receives only the raw, restricted sensor data for a given segment and must generate the associated Ground Truth (GT) Daily Drilling Report (DDR) text. More details are provided in Appendix A.

### 4.2. Metrics

Evaluating DDR generation requires assessing both contextual correctness and numerical fidelity. We employ a structured LLM-as-a-judge protocol utilizing GPT-5 (5).

The automatic evaluator compares each generated entry against its reference across distinct criteria, which are grouped into four weighted dimensions: (i) primary operation correctness (A), (ii) numerical fidelity (B: Depth Match, E: Parameters Match, F: Equipment Dimensions), (iii) operational detail coverage (D: Secondary Operations, G: BHA Type, H: Operational Details), and (iv) stylistic alignment (C: Conciseness). See Appendix B for more details.

## 5. Results

### 5.1. Quantitative Evaluation

We evaluate three paradigms—a VLM, an Agentic framework, and a Multimodal TSFM (MM-TSFM)—across nine operational classes (Table 1; Appendix G). The VLM establishes a lower bound (55.4%), reflecting an expected "visual translation bottleneck": rendering long, high-frequency sequences as static plots attenuates fine-grained numerical cues. We read this as a quantification of the visual paradigm under standard plotting, not a like-for-like comparison against direct-access methods. The GPT-5 Agentic framework scores highest (70.1%); its iterative reasoning (averaging 7 LLM calls per generation) excels at precise numerical extraction and at rare, high-complexity classes such as Coring. The MM-TSFM narrows the gap by preserving multivariate signal behavior—its frozen Chronos-2 backbone reaches 67.5%, beating lighter CNN variants—and notably surpasses the Agentic approach on the well-construction phases Cementing (65.1%) and Casing (58.9%).

These gains come far cheaper. The single-pass MM-TSFM architectures cut inference cost by over two orders of magnitude relative to the Agentic approach's iterative calls: Chronos-2 averages 1.58 s and $0.00126 per call. The CNN variant is marginally cheaper ($0.00112), but Chronos-2's accuracy gains justify the difference, making it the most

viable candidate for near real-time industrial reporting.

## 5.2. Qualitative Analysis

The three architectures show complementary strengths (Figure 2; full breakdown in Tables 15 and 16). We use *temporally extended contextual events* for operations identified by a morphological signature spanning the full segment—e.g., the stepwise hookload increase of a casing run or the sustained no-flow, no-rotation state of a flow check—rather than by any single timestamp. Such events are scored by the recognition rubrics A (Primary Operation), G (BHA Type), and H (Operational Details), as distinct from the point-numerical rubrics B (Depth) and E (Parameters).

This contrast maps onto the architectures. The Agentic system leads the point-numerical rubrics, recovering exact telemetry such as 585 GPM/2744 PSI (Cat. B: 97%, Cat. E: 87%). The TSFM leads the equipment- and detail-oriented rubrics (Cat. G: 71%, Cat. H: 58%) and, crucially for the class-level edge above, correctly names the operation in these well-construction phases where the agent defaults to generic circulating or tripping (Figure 3). The VLM, weakest at numerical grounding, still translates visually distinct step-changes into broad procedural narratives.

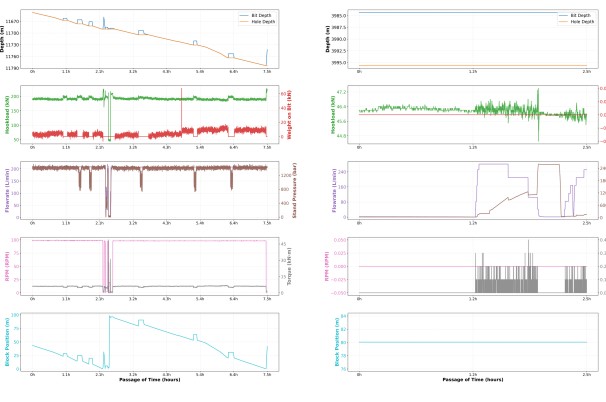

**Generated Entries vs. Reference**

**Reference:** DRILLED 6" W/ BHA (RSS + M/LWD) F/ 11647' T/ 11783' - 220 GPM/1400 PSI, 100 RPM, 5-7 KFTLB.

**VLM:** DRILLING F/ 11,670' T/ 11,790'. FR ~220 GPM, RPM ~100, SPP ~1200 PSI, TQ ~6–8 KFT-LB, HL ~190–205 KLB. Mid-run pumps-off/connection: TQ spike ~45 KFT-LB, SPP to ~0, then normalized. Flow-out steady when pumps on.

**Agentic:** ROTARY DRILL HOLE W/ ROTARY BHA F/ 11,647' T/ 11,783'. 208 GPM/1411 PSI, 98 RPM.

**TSFM:** DRILL 6" HOLE W/ RSS BHA F/11612' T/11977'.

*Figure 2.* Visual sensor data and corresponding DDR generation examples for a drilling operation across approaches.

**Generated Entries vs. Reference**

**Reference:** CONT'D 9-5/8" CSG **CMT JOB** AS PER PROGRAM.

**VLM:** CIRCULATE HOLE STATIC; INTERMITTENT ROTATION OFF-BTM. FLOW 0→~250 GPM; SPP BUILT TO ~1200 PSI THEN DOWN. STANDPIPE PRESSURE TEST W/ PUMPS OFF TO ~2400 PSI. RESUMED LOW-RATE CIRC. WOB ZERO; BIT/HOOK DEPTH CONSTANT. PIT VOL GAIN THEN BLEED-OFF. NO OVERPULL/SLACK-OFF OR LOSSES EVIDENT.

**Agentic:** CIRCULATED HOLE CLEAN 226 GPM/743 PSI. PRESSURE TEST STANDPIPE TO 2,600 PSI.

**TSFM:** PERFORMED 2ND STAGE **CMT JOB** AS PER PROGRAM.

*Figure 3.* Visual sensor data and text examples highlighting TSFM's contextual recognition of casing cementing.

## 5.3. Ablation Studies

We conducted ablations to isolate the architectural choices and prompting strategies. (Table 2).

For the MM-TSFM framework, a multi-stage training routine utilizing a frozen high-capacity temporal backbone (Chronos-2) combined with a label-grounded output format yielded the most significant improvements (+6.3% and +3.3%, respectively). Conversely, over-specifying the prompt degraded performance, as excessive instruction caused the LLM with sub-7B parameters to prioritize redundant linguistic noise over the nuanced interpretation of sensor latents. For the Agentic and VLM approaches, increasing inference-time reasoning budgets and utilizing few-shot random sampling provided marginal to moderate improvements. Comprehensive details are provided in Appendices I, K, and J.

*Table 2.* Summary of performance gains from key architectural and prompting optimizations.

| Paradigm | Optimization Feature | $\Delta$ Gain (%) |
|----------|---------------------|------------------|
| **MM-TSFM** | Multi-stage Training + Frozen TS Backbone | +6.3 |
| | Label-Grounded Output Format | +3.3 |
| | Text Decoder Prompt Refinement | +1.7 |
| **Agentic** | High-Thinking Reasoning Budget | +0.5 |
| **VLM** | 9-Shot Sampling | +1.6 |

## 6. Conclusion and Future Directions

We demonstrated that robust DDR generation requires complementary reasoning strategies. Our central empirical finding is a clear functional division of labor: Agentic models excel at verifiable numerical extraction, while MM-TSFM architectures better identify complex operational states at far lower inference cost—a complementarity that follows directly from our comparison and does not hinge on any single system being uniformly best.

This motivates hybrid systems, which we leave to future work. A concrete instantiation exposes the pre-trained TSFM as a tool within the Agentic ReAct loop: its fast, single-pass estimate of operational context and BHA metadata would route the agent's subsequent SQL queries before it commits tool-verified values, inheriting the TSFM's contextual priors at the cost of one extra forward pass.

Two limitations bound our scope. We evaluate the paradigms in isolation rather than validating a combined system, and our VLM baseline encodes telemetry as static 2D plots—an inherently lossy representation that characterizes the visual paradigm rather than upper-bounding it. Higher-fidelity encodings (multi-resolution, paginated, or table-augmented) are a promising way to strengthen it.

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

# Appendix

## A. Dataset Details

### A.1. Data Composition and Partitioning

The large-scale industrial dataset is collected from multiple wells and comprises over 3,078 days of continuous telemetry. The dataset contains 22,483 distinct operational events, nearly 2.9 billion sensor datapoints, and 277,000 words. These events encompass nine distinct operational classes: Completion (CM), Cementing (CMT), Coring (CORE), Casing (CSG), Drilling (DRILL), Drill Out (DRLOUT), Reaming (REAM), Stuck Pipe (STKP), and Tripping (TRIP). Each sample consists of multivariate sensor data $\mathbf{X} \in \mathbb{R}^{T \times C}$ with $C = 11$ channels sampled at 1 Hz. Operational segments are provided as temporal windows $\mathbf{X}_i \in \mathbb{R}^{T_i \times C}$ extracted from industrial pipelines. No categorical activity labels are provided; instead, the information for each segment is restricted to raw sensor telemetry and its associated Ground Truth (GT) Daily Drilling Report (DDR).

To ensure generalization and prevent temporal leakage, the data is partitioned using a two-stage 80/20 split: the total dataset was first divided into a training set (80%) and a hold-out test set (20%), after which the training set was further partitioned into training and validation subsets using the same 80/20 ratio. Table 4 summarizes the distribution of samples, temporal lengths ($T$), and DDR word lengths across these splits.

### A.2. Sensor Channel Definitions

The input data captures 11 key surface and downhole channels essential for interpreting drilling dynamics. These channels are categorized by their physical domains in Table 3.

*Table 3.* Description of Input Sensor Channels ($C = 11$)

| Category | Channel Name | Description |
|---|---|---|
| Positional | Bit Depth | The measured depth of the drill bit face within the wellbore. |
| | Hole Depth | The total depth reached by the wellbore. |
| | Block Position | Vertical position of the traveling block assembly; used to track pipe movement and velocity. |
| Mechanical | Hook Load | Total weight suspended from the hook, including the drill string and traveling assembly. |
| | Weight on Bit | The drill string weight applied to the bit, calculated as total string weight minus hook load. |
| | Rotary Speed | The rotational velocity of the drilling assembly measured in revolutions per minute (RPM). |
| | Torque | The rotational force/resistance encountered by the drill string during operation. |
| Hydraulic | Standpipe Pressure | The fluid pressure inside the drill string required to maintain circulation. |
| | Flow Rate | The rate at which drilling fluid is pumped into the well. |
| | Flow Out (%) | The percentage of fluid returning from the wellbore relative to maximum return capacity. |
| | Tank Volume | Total volume of drilling fluid stored in the active surface pits; critical for well control. |

### A.3. Sequence and Report Complexity

Each segment is associated with a DDR containing contextual descriptions and numerical values. Across the dataset, segment lengths vary significantly, with a minimum length of 21 and a maximum of 86,401, while the average length is 11,830.50. The DDRs vary in complexity, with word lengths ranging from 1 to 280 words and an overall mean length of 12.32 words. This variance requires the architecture to handle both high-frequency signal processing and long-range linguistic dependencies.

*Table 4.* Data Statistics: Sequence Length ($T$) and DDR Word Length

| Split | Samples | Sequence Length ($T$) | | | | GT Caption (DDR) Word Length | | | |
|---|---|---|---|---|---|---|---|---|---|
| | | Min | Max | Mean | Median | Min | Max | Mean | Median |
| Train | 13,788 | 21 | 86,401 | 11,672.4 | 7,201 | 1 | 278 | 12.59 | 10.0 |
| Validation | 4,073 | 32 | 86,400 | 12,500.2 | 9,001 | 1 | 280 | 12.15 | 10.0 |
| Test | 4,622 | 69 | 86,401 | 11,712.0 | 7,201 | 1 | 212 | 11.67 | 10.0 |
| Overall | 22,483 | 21 | 86,401 | 11,830.5 | 7,201 | 1 | 280 | 12.32 | 10.0 |

## B. LLM-as-a-Judge Evaluation Criteria and Weighting

To rigorously assess both the semantic correctness and factual fidelity of the generated DDRs, we employ a structured, multi-criteria LLM-as-a-judge protocol. The evaluator LLM independently scores each prediction against a ground-truth reference across eight distinct criteria. To prevent the model from internally mis-weighting these dimensions during a single-pass generation, we extract binary judgments (0 or 1) for each criterion and apply an external aggregation schema (5).

The criteria and their respective weights are distributed as follows:

*Table 5.* LLM-based Evaluation Rubric for DDR Generation

| ID | Criterion | Description | Weight |
|----|-----------|-------------|--------|
| A | Primary Operation | Evaluates if the main semantic operation (e.g., drilling, tripping, circulating) matches the reference or provided code. | 30% |
| B | Depth Match | Verifies that depths or depth ranges fall within an acceptable operational tolerance. | 15% |
| C | Conciseness | Penalizes unnecessary repetition or filler, enforcing a telegraphic rig manager style. | 2% |
| D | Secondary Operations | Ensures all secondary or parallel operations present in the reference are captured. | 15% |
| E | Parameters Match | Verifies numeric telemetry parameters (flow, torque, RPM, etc.) match within a $\pm 10\%$ tolerance. | 15% |
| F | Equipment Dimensions | Checks that specified equipment and hole dimensions align within tolerance. | 5% |
| G | BHA Type | Confirms specific Bottom Hole Assembly designations are correctly identified. | 3% |
| H | Operational Details | Captures domain specific anomalies like tight holes, overpulls, or fluid losses. | 15% |

**Rationale for Rubric Weighting.**    The weighting schema reflects the strict requirements of industrial reporting, where factual inaccuracies carry severe operational consequences. The heaviest weights are assigned to the Primary Operation (30%) and critical numeric/state anchors like Depths (15%), Parameters (15%), and All Operations (15%). Together, these constitute 75% of the final score, ensuring that a model cannot achieve a high rating simply by generating fluent but hallucinated text. Conversely, stylistic elements like Conciseness (2%) and granular equipment sub-classifications like BHA Type (3%) receive minimal weighting. While these elements contribute to the professional formatting of the report, failing to capture them does not fundamentally misrepresent the physical state of the wellbore.

# C. Extended Methodology: Multimodal TSFM Architecture

**Architecture Overview**    We adapt the *OpenTSLM-Flamingo* (12) architecture, a multimodal framework designed to bridge the gap between continuous time-series telemetry and discrete linguistic reasoning. Our architecture, shown in Figure 1c, consists of three primary components: a high-capacity Time-Series Encoder backbone, a Perceiver Resampler for dimensionality bottlenecking, and a Large Language Model (LLM) modified with interleaved Gated Cross-Attention (GCA) layers serving as the text decoder. Unlike standard "prefix-tuning" methods where sensor features are prepended to the text prompt, the Flamingo-based approach keeps the time-series and text representations distinct until the cross-attention stage. This prevents the LLM's context window from being consumed by sensor tokens, enabling the model to handle significantly longer drilling intervals without hitting token limits.

**Foundational TS Encoder Backbone**    A key departure from the original *OpenTSLM* implementation is our replacement of the default patch-based CNN with specialized Time-Series Foundation Models (TSFMs). While the baseline framework utilizes a generic patch encoder to process 1D signals, we evaluate the impact of high-capacity pre-trained backbones, specifically **Chronos-2** and **Toto** (7), that possess strong temporal and morphological priors. The TSFM backbone produces a sequence of latent vectors $\mathbf{Z}_{raw}$, which the Perceiver Resampler compresses into a fixed-length set of temporal tokens $\mathbf{X}_{ts} \in \mathbb{R}^{N \times d}$, effectively distilling high-frequency sensor data into a format digestible by the LLM.

**Gated Cross-Attention Mechanism**    To integrate temporal tokens $\mathbf{X}_{ts}$ into the LLM decoder (Gemma3-1B), we utilize the GCA mechanism as established in the Flamingo architecture (1) and adapted for time-series in (12). This approach interleaves GCA layers between existing LLM blocks, where the integration is governed by the following operation:

$$\mathbf{h}'_{text} = \mathbf{h}_{text} + \tanh(\alpha) \cdot \text{CrossAttention}(\mathbf{h}_{text}, \mathbf{X}_{ts}) \tag{1}$$

where $\mathbf{h}_{text}$ represents the linguistic hidden states and $\alpha$ is a learnable gating parameter. Following the initialization strategy proposed by (1), $\alpha$ is initialized to zero; this ensures that at the onset of training, the model's output remains identical to the original LLM, gradually incorporating sensor features as the gating parameter is optimized. This mechanism allows the model to dynamically attend to multivariate sensor data conditioned on the text prompt without disrupting the pre-trained linguistic weights. In line with the parameter-efficient philosophy of this framework, we freeze the original transformer blocks and restrict training to the GCA layers of the text decoder, thereby preserving the LLM's foundational reasoning capabilities while adapting it to the drilling domain.

**Multi-stage Training Strategy**    We employ a two-stage optimization routine to align continuous sensor latents with the discrete requirements of industrial reporting.

- *Stage 1: Multi-Scale Operational Pre-training.* This phase establishes operational context through two parallel classification routines: (i) **Global Alignment:** The entire architecture is trained on activity classification to pre-condition the gated cross-attention layers and perceiver resampler to sensor inputs. (ii) **TS Encoder Backbone Refinement:** The encoder backbone is trained independently with a classification head to ensure high-fidelity feature extraction.

- *Stage 2: Label-Grounded Captioning.* For DDR generation, we initialize the model using weights from the *Global Alignment* routine. Crucially, we then re-initialize the TSFM backbone by overwriting its weights with those from the *Backbone Refinement* routine. During fine-tuning, the TSFM backbone remains frozen, acting as a stable feature extractor. We employ a grounded output format: ⟨*Class label*⟩ *DDR:* ⟨*DDR*⟩, which anchors the latent representations in the operational context prior to DDR generation.

As detailed in Table-2, this grounding mechanism ensures that the latent representation is anchored in the correct operational context before narrative generation begins. The efficacy of freezing the backbone versus full fine-tuning is further detailed in the ablation studies (Section I.2).

## D. Extended Methodology: Vision-Language Approach

**Visual Encoding**   For each operational segment, the raw 1 Hz sensor channels are explicitly rendered into a 2D multi-panel plot, with each signal visualized in a separate subplot to preserve channel-specific morphology (see Figure 4). The resulting figures are base64-encoded and passed to the VLM alongside a structured system prompt. This approach relies on the VLM's ability to extract numerical parameters and operational logic directly from visual trends rather than raw numerical sequences.

**Prompting Strategy and Few-Shot Setup**   As DDR generation requires a telegraphic and concise style, we designed a system prompt containing an explicit anti-hallucination constraint—instructing the model to only extract numerical parameters that are definitively legible on the plot's text, legends, or axes. The prompt also includes reference examples organized by operation type (e.g., cementing, casing, drilling) to enforce the standard rig manager reporting style. The exact prompts utilized are provided in Section F.2.

While the system prompt provides general stylistic guidance, we also evaluated augmenting the conversational context with grounded few-shot image-text pairs. In these configurations, we appended 3 or 9 image-text (ground truth) pairs after the system prompt to better anchor the final response. The quantitative impact of these few-shot strategies is detailed in the ablation study (Section J).

## E. Extended Methodology: Agentic Approach

**Agent loop.**   Given a segment $a_i = (t_i^{\text{start}}, t_i^{\text{end}}, y_i)$, the agent executes a ReAct-style (23) tool-calling loop for at most $K=100$ iterations, after which it synthesizes a final answer from gathered evidence.

**Toolset.**   The toolset provides two classes of access. *SQL-based access* via `query_dataframe_view` exposes the full sensor window as a DuckDB table, enabling window aggregates, filters, and multi-step analytical pipelines. *Signal-level primitives* offer higher-level operations: distributional summaries, piece-wise trend fitting, depth summaries, anomaly detection, cross-sensor comparison, derived signal transforms, and time-series visualization returned as base64 images for direct inspection.

**Anti-hallucination constraint.**   A key design principle is that all numerical values in the final DDR must be explicitly verified by at least one tool output. The system prompt enforces this constraint explicitly and distinguishes it from equipment-type inferences, which the model is permitted to make from operational context when confident.

**Relation to input formulation.**   The Agentic approach realizes $f_\theta \colon (\mathbf{X}, \mathcal{A}) \to \mathbf{Y}$ by conditioning each generation step on the activity segment $a_i$ and on-demand sensor query outputs. Unlike the multimodal architecture, no fixed-length embedding of $\mathbf{X}_i$ is constructed; instead, the model selects and aggregates evidence iteratively, focusing retrieval on the sub-windows and channels most relevant to the inferred operation type.

# F. Prompting Details

## F.1. Multimodal TSFM Prompting Details

This section provides the full prompt templates discussed in Section I.6. To integrate time-series modalities into the autoregressive framework of the language model, we utilize the conditioning strategy outlined by OpenTSLM (12). Building upon multimodal techniques originally designed for vision-language tasks, this method employs learnable special tokens—specifically `<TS>` and `<|endofchunk|>`—to facilitate the flexible interleaving of textual and temporal data sequences. When the model encounters a `<TS>` token, it is triggered to incorporate the corresponding time-series latent embeddings through a gated cross-attention mechanism. The `<|endofchunk|>` token is then used to explicitly mark the conclusion of the textual description associated with that specific signal.

To illustrate how these structural elements are applied, we present both the descriptive and concise prompt templates below. A sample of the model predictions for identical sensor data sequences is provided within each template block to demonstrate the typical variations in output behavior.

---

**Drilling DDR Captioning Input Prompt (Descriptive)**

```
You are an expert drilling engineer analyzing multivariate sensor data from 11 sensors.  Your task is
two-fold:
1.  Classify the drilling operation being performed.
2.  Generate a concise Daily Drilling Report (DDR) entry describing the operation.

POSSIBLE OPERATIONS AND DEFINITIONS:
- CM: Preparation, running, landing, and testing of a new well completion.
- CMT: Cementing operations including rigging up/down lines.
- CORE: Coring operations including tripping and circulating.
- CSG: Running casing and liners.
- DRILL: Drilling new hole, including hole opening and under-reaming.
- DRLOUT: Drilling out cement/shoe and new formation for LOT/FIT.
- REAM: Reaming or washing down to address hole conditions (excludes under-reaming).
- STKP: Remedial actions to free stuck pipe.
- TRIP: Tripping pipe (POOH/RIH) and BHA handling.

<TS> The following is the rotational velocity (RPM) data, it has mean 0.0054 and std 0.0078 <|endofchunk|>
<TS> The following is the standpipe pressure (PSI) data, it has mean 0.0000 and std 0.0000 <|endofchunk|>
<TS> The following is the weight on bit (klbs) data, it has mean 0.0000 and std 0.0000 <|endofchunk|>
<TS> The following is the torque (klb-ft) data, it has mean 0.0143 and std 0.0709 <|endofchunk|>
<TS> The following is the flow rate (GPM) data, it has mean 0.0000 and std 0.0000 <|endofchunk|>
<TS> The following is the flow out percentage (%) data, it has mean 0.0500 and std 0.0001 <|endofchunk|>
<TS> The following is the bit depth (feet) data, it has mean 761.0937 and std 0.0000 <|endofchunk|>
<TS> The following is the block position (feet) data, it has mean 64.8073 and std 27.0327 <|endofchunk|>
<TS> The following is the hook load (klbs) data, it has mean 46.1022 and std 1.8282 <|endofchunk|>
<TS> The following is the hole depth (feet) data, it has mean 8938.0010 and std 0.0000 <|endofchunk|>
<TS> The following is the tank volume (barrels) data, it has mean 454.5240 and std 1.1347 <|endofchunk|>

Based on the drilling sensor data provided above, respond with the operation label followed by a DDR entry.
Format your response EXACTLY as:
<LABEL> DDR: <description>

________________________________________________________________________

GROUND TRUTH
1.  TRIP DDR: RIH W/ 6-1/8" RSS BHA ON 5-1/2" DP F/ 10,295' TO 11,999'.
2.  TRIP DDR: RIH W/ 8 1/2" RSS BHA F/7,250' TO 11,440'

MODEL PREDICTIONS
1.  TRIP DDR: RIH 6in RSS BHA ON 5in DP SINGLES TO 13,315' FT. FILLED BHA EVERY 3 HOURS.
2.  TRIP DDR: RIH 6'' RSS / TRIPLE COMBO BHA FROM 8063' TO 11447'

________________________________________________________________________

Notation:  The special tokens <TS> and <|endofchunk|> denote the start and end of the time-series-related
text, respectively.
```

---

**Drilling DDR Captioning Input Prompt (Concise)**

```
Context:  Drilling Sensor Analysis.
Allowed Labels:  CM, CMT, CORE, CSG, DRILL, DRLOUT, REAM, STKP, TRIP.

<TS> The following is the rotational velocity (RPM) data, it has mean 0.0054 and std 0.0078 <|endofchunk|>
<TS> The following is the standpipe pressure (PSI) data, it has mean 0.0000 and std 0.0000 <|endofchunk|>
<TS> The following is the weight on bit (klbs) data, it has mean 0.0000 and std 0.0000 <|endofchunk|>
```

```
<TS> The following is the torque (klb-ft) data, it has mean 0.0143 and std 0.0709 <|endofchunk|>
<TS> The following is the flow rate (GPM) data, it has mean 0.0000 and std 0.0000 <|endofchunk|>
<TS> The following is the flow out percentage (%) data, it has mean 0.0500 and std 0.0001 <|endofchunk|>
<TS> The following is the bit depth (feet) data, it has mean 761.0937 and std 0.0000 <|endofchunk|>
<TS> The following is the block position (feet) data, it has mean 64.8073 and std 27.0327 <|endofchunk|>
<TS> The following is the hook load (klbs) data, it has mean 46.1022 and std 1.8282 <|endofchunk|>
<TS> The following is the hole depth (feet) data, it has mean 8938.0010 and std 0.0000 <|endofchunk|>
<TS> The following is the tank volume (barrels) data, it has mean 454.5240 and std 1.1347 <|endofchunk|>

Task:  Provide Operation Label and DDR description.
Constraint:  Use EXACT format below.  No preamble.
Format:  [LABEL] DDR: [description]
Answer:
________________________________________________________________

GROUND TRUTH
1.  TRIP DDR: RIH W/ 6-1/8" RSS BHA ON 5-1/2" DP F/ 10,295' TO 11,999'.
2.  TRIP DDR: RIH W/ 8 1/2" RSS BHA F/7,250' TO 11,440'

MODEL PREDICTIONS
1.  TRIP DDR: RIH W/ 6" RSS/LWD BHA ON 4" DP STDS TO 12100'.
2.  TRIP DDR: RIH W/ 8.5" RSS/LWD BHA TO 10906'
```

While both prompts enable the model to successfully classify the overarching operations, qualitative comparisons of the resulting text suggest that the concise prompt encourages tighter alignment with the raw telemetry data. For instance, in the first example, the concise setting provides a closer depth approximation (12,100 ft relative to the 11,999 ft ground truth) and avoids the extraneous procedural details generated by the descriptive prompt. In the second example, a minor trade-off emerges: the concise prompt correctly identifies the 8.5" BHA diameter—avoiding the descriptive prompt's hallucination of a 6" diameter—although the descriptive prompt yields a slightly more precise depth estimation. Ultimately, because the concise prompt systematically mitigates these minor hallucinations and provides marginally more robust results across a wider range of settings, we finalized it as the standard template for our experiments.

## F.2. VLM Prompting Details

The Vision-Language Model (GPT-5) was prompted to act as a domain expert with strict constraints on numerical hallucination. Figure 4 shows an example of the base64 image used for the VLM approach, while the full system prompt and the standard user query are detailed in the blocks below.

---

**VLM System Prompt**

```
You are an expert Oil & Gas Drilling Engineer.  Your task is to analyze drilling sensor plot images and
translate the visual data into a standard Daily Drilling Report (DDR) string.

Write the report exactly as a rig manager would:  telegraphic, ultra-concise, and completely devoid of
conversational filler.

Carefully examine the plot and construct your summary based on these operational pillars:
1.  Primary Activity:  Identify the overarching operation (e.g., DRILLING, RIH, POOH, CIRC, RUN CSG)...
2.  Key Parameters:  Note start/end depths, flow rates (gpm), RPM, torque, hook load, or mud weight.
3.  Anomalies:  Identify visual markers for anomalies...

***STRICT ANTI-HALLUCINATION RULE***
You must only include specific numerical parameters if they are definitively legible in the plot's text,
legends, or axes.  Do not infer, guess, or assume standard industry values...

________________________________________________________________

REFERENCE EXAMPLES BY OPERATION TYPE
[Note:  The prompt includes 40+ reference examples across 9 categories; a subset is shown below for
brevity.]

Drilling:
- RSS Drill 12 1/4" hole F/ 5,061 ft to 5,227 ft.  Flow= 728 gpm, WOB= 18 klbs, SPP= 1808 psi, RPM= 114, ROP=
66.2 fph.  No losses.
- DOWN LNK, RSS DRILLED 12 1/4" HOLE FROM 4,985 FT TO 5,063 FT. WASH UP & REAM DOWN EVERY STAND.

Tripping:
- POOH with 12 1/4" RSS assembly from 7860 ft to 13 3/8" CSG shoe @ 4,868 ft.  Well taking correct metal
displacement.
```

**VLM User Prompt**

Based strictly on the visual evidence in the attached plot, generate the corresponding DDR string. Focus on the primary activity, key parameters, and anomalies. Output ONLY the string (no explanations or additional text).

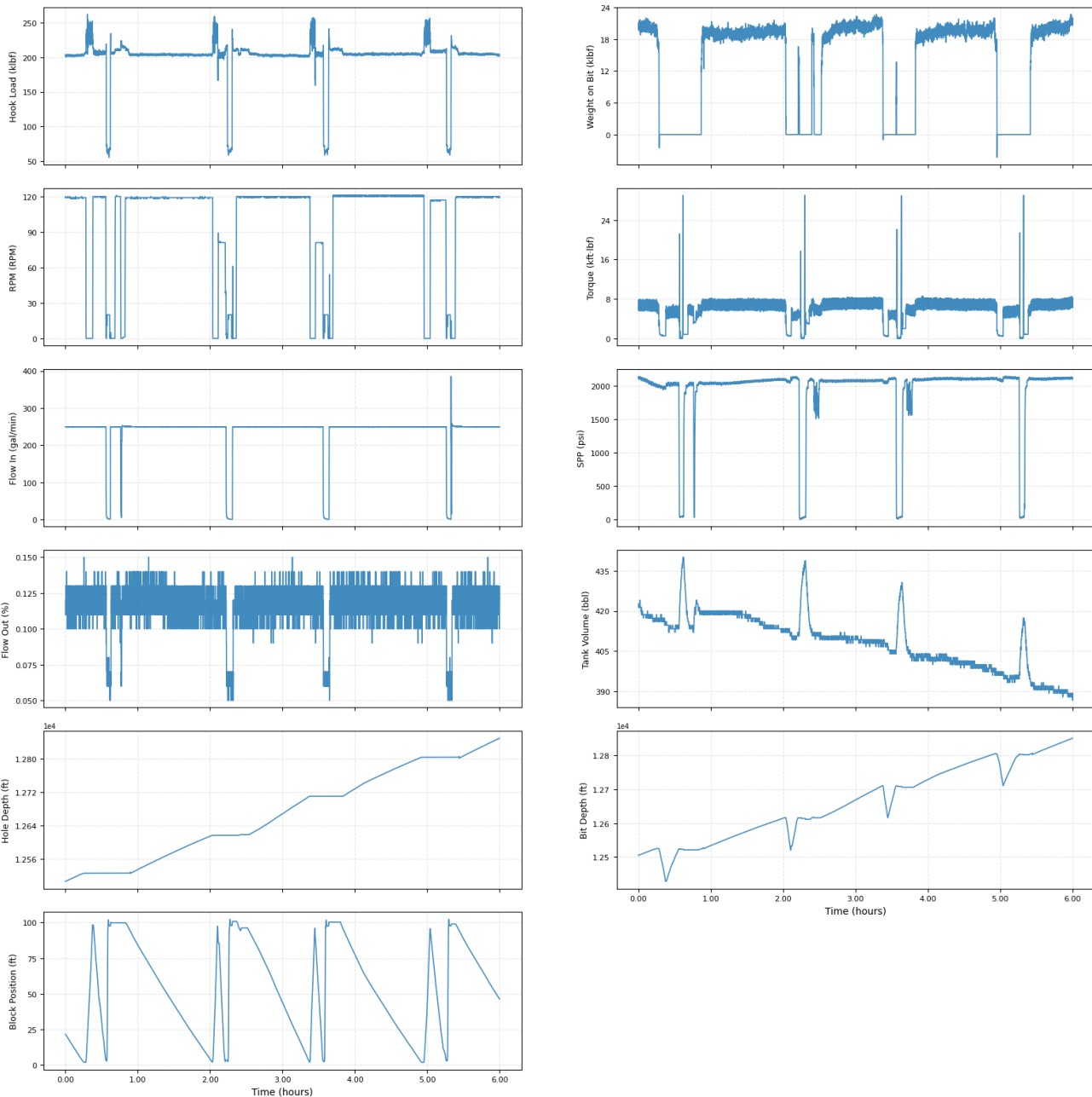

*Figure 4.* Illustrative 6-hour window of multivariate drilling telemetry used for the VLM approach. The subplots visualize 11 key sensor channels capturing cyclic drilling sequences.

## F.3. Agentic Approach Prompting Details

The Agentic framework (GPT-5) uses a ReAct-style loop. The full system prompt is constructed by concatenating the Domain Prompt, the Tool-Use Policy, and (in the Expert configuration) the Diagnostic Playbook.

### F.3.1. BASE DOMAIN PROMPT

This section defines the core persona, anti-hallucination rules, and telegraphic style requirements.

---

**Agentic: Base Domain Prompt**

```
You are an expert Oil and Gas Drilling Engineer.  Your task is to analyze drilling sensor data and generate
a standard Daily Drilling Report (DDR) string.

Write exactly as a rig manager would:  telegraphic, ultra-concise, no conversational filler...

ANTI-HALLUCINATION RULE:
- Numerical measurements must be verified by tool outputs.  Do not fabricate measurements.
- Hole size, BHA type, and equipment type are standard DDR fields...  Include them when you can confidently
identify them from the sensor evidence.

Equipment terminology:
- Casing runs:  "RIH [size] CSG" or "RUN [size] CSG" -- NOT "RIH W/ BHA"
- Drill-out:  "DRILL OUT CMT" or "D/O CMT/SHOE/PLUGS" -- NOT "REAM" or "WASH"

[Reference examples follow, identical in style to the VLM examples but including specific Agentic
tool-verified parameters.]
```

---

### F.3.2. EXPERT DIAGNOSTIC PLAYBOOK

This prompt was added in the **Expert Prompt** ablation (Table 5) to provide the model with "signatures" for operations that are frequently misclassified.

---

**Expert Diagnostic Playbook (Ablation)**

```
EXPERT DIAGNOSTIC PLAYBOOK
The following playbook concentrates knowledge on the operations the model is most likely to misclassify.

=== CASING / LINER RUN vs DRILL-STRING TRIP ===
Primary risk:  defaulting to "RIH W/ BHA" when the reference is "RIH 9-5/8" CSG".
1.  NO rotation.  Casing/liner runs are NEVER rotated.
2.  HOOK-LOAD STAIRCASE (the single strongest tell).  Casing strings add WEIGHT at surface every joint.
3.  FILLING THE STRING: Signature:  short, repeated pump BURSTS (100-400 GPM) for 1-3 min while block_pos
barely moves.

=== FLOW CHECK vs TRIP / CIRCULATION ===
Signature:  Pumps OFF (< 5 GPM), No rotation, Block not moving, Bit depth constant for >= 3 min.

=== STUCK PIPE / WORKING STRING ===
Signatures:  Bit depth constant >= 5 min AND hook load cycling (Overpull/Slack-off).  Jarring signals.
```

---

### F.3.3. TOOL-USE POLICY

This block governs how the model interacts with the DuckDB/SQL environment and the final formatting of its answer.

---

**Tool-Use & Output Policy**

```
Analysis strategy:
- First identify the operation type, then gather parameters relevant to it.
- Use your drilling expertise to interpret what the sensor data means -- do not just report raw numbers.
- Run an operation-parity check:  keep only operations that are central and strongly evidenced.

Tool-use policy:
- Use query_dataframe_view with SQL (DuckDB dialect) for filtering, aggregation, and windowing.
- Use direct sensor tools when you need exact values.
- Think by gathering evidence, but do not reveal chain-of-thought in the final answer.

Final answer rules:
- Respond with the DDR string only.  No explanations or labels.
```

---

# G. Extended Quantitative Results

This section provides the granular quantitative evaluations supporting the main findings in Section 5.1.

### G.1. Pairwise Model Comparisons

As illustrated in the pairwise performance heatmaps (Figure 5), the Agentic approach consistently outperforms both baselines. The VLM fails in roughly half of the cases where either the Agentic or Multimodal TSFM succeeds. The margin between the Agentic and Multimodal TSFM models is tighter, with the Agentic model holding a slight edge (outperforming the TSFM in 17.6% of cases compared to a 12.1% win rate for the TSFM), highlighting their overlapping but distinct capabilities in handling complex telemetry.

*Figure 5.* Pairwise model performance distribution across the test set. "Good" corresponds to an LLM-eval score $\geq 70\%$. The heatmaps illustrate the high success overlap (60.7%) between the Agentic and TSLM models.

# H. Inference Cost and Latency Details of Multimodal TSFM

The inference cost for the Multimodal TSFM architecture was calculated based on the on-demand pricing of an NVIDIA V100 GPU across major cloud providers. To reflect a flexible deployment scenario, we utilized standard hourly rates from Google Cloud Platform (GCP), Amazon Web Services (AWS), and Microsoft Azure without factoring in monthly or yearly commitment discounts. The average cost across these providers is approximately $2.87 per hour. This average rate was used to derive the per-call costs detailed in Table 6, accounting for the model throughput and token generation rate on this hardware configuration.

**Inference Latency**    In addition to cost efficiency, we evaluated the inference latency of the MM-TSFM architectures. Because the processing speed of time-series models is dependent on the input sequence length provided in the batch, we measured the average inference time over a subset of 200 samples from the test set. The window lengths for these samples ranged from 92 to 2588 data points, with an average length of 742.6 points.

As shown in Table 6, the CNN (Default) configuration provides the most efficient baseline with an average inference time of 1.4029 seconds and a per-call cost of $0.00112. However, the Chronos-2 variant remains highly competitive, requiring only a marginal cost increase (approximately $0.00014 additional per call) and a slight latency trade-off. Given the significant performance improvements in classification and captioning tasks observed with the Chronos-2 backbone, it represents the most balanced candidate for high-accuracy industrial reporting where the small overhead is justified by the gain in predictive quality.

*Table 6.* Average Inference Time and Cost for MM-TSFM Architectures

| TS Encoder Backbone | Language Model | Inference Time (s) | Inference Cost (USD) |
|---|---|---|---|
| CNN (Default) | TSLM - Gemma3-1B | 1.4029 | 0.00112 |
| Chronos-2 | TSLM - Gemma3-1B | 1.5769 | 0.00126 |
| Toto | TSLM - Gemma3-1B | 1.6163 | 0.00129 |

# I. Multimodal TSFM Ablation Studies

## I.1. Foundation Backbones

We evaluate the standalone classification performance of the three time-series encoder backbones prior to their integration into the multimodal framework. As detailed in Table 8, the specialized TSFMs (Chronos-2 and Toto) significantly outperform the baseline Patch-based CNN, with Chronos-2 achieving a peak standalone accuracy of 78.16%.

However, the critical evaluation lies in their robustness when integrated into the broader OpenTSLM-Flamingo architecture (Table 7). While standalone accuracy is a strong indicator of feature extraction capability, the multimodal setting introduces a dimensionality bottleneck via the Perceiver Resampler and alignment constraints through the Gated Cross-Attention mechanism. In this integrated environment, Chronos-2 maintains its performance advantage. This demonstrates that its learned temporal representations are not only highly accurate but also effectively compressible and structurally compatible for cross-attention with the linguistic decoder, making it the most viable backbone for the end-to-end reporting task.

## I.2. Multi-Stage Training Strategy

Our ablation results indicate that a single-pass fine-tuning approach is insufficient for mapping complex industrial telemetry to natural language. Instead, we utilize a two-stage optimization routine to ensure both architectural alignment and domain-specific feature stability.

**Stage 1: Operational Alignment and Refinement**

In this stage, we address the modality gap through two parallel routines:

*Global Alignment*: We train the full multimodal stack on activity classification. This initial phase optimizes the Perceiver Resampler and Gated Cross-Attention (GCA) layers, conditioning the model to attend to temporal tokens. Rather than projecting latents directly into the LLM's vector space, this routine trains the gating parameters and cross-attention weights, allowing the frozen linguistic model to dynamically utilize continuous sensor features.

*Table 7.* Stage-1: Multimodal TSFM Performance on Activity Label Classification

| Model | Initial Weights | TS Encoder Train Mode | Classification Accuracy (%) |
|---|---|---|---|
| OpenTSLM CNN (Default) | Published Weights | Full finetune | 64.75 |
| OpenTSLM Chronos-2 | Published Weights | Full finetune | **70.98** |
| OpenTSLM Toto | Published Weights | Full finetune | 67.59 |

*Backbone Refinement*: Simultaneously, the TSFM backbone is trained independently with a classification head. This step refines the initial, generalized foundation weights to capture the specific high-frequency signatures and operational nuances of drilling telemetry.

*Table 8.* Stage-1: TS Encoder Backbone Performance on Activity Label Classification

| TS Encoder Backbone | Initial Weights | Train Mode | Classification Accuracy (%) |
|---|---|---|---|
| CNN (Default) | Random | Full finetune | 62.84 |
| Chronos-2 | Published Weights | Full finetune | **78.16** |
| Toto | Published Weights | Full finetune | 76.60 |

**Stage 2: Label-Grounded Captioning**

For the final DDR generation task, the multimodal architecture is initialized using weights from the Global Alignment routine. Crucially, the TSFM backbone is re-initialized with the domain-refined weights from Stage 1 and subsequently frozen.

As demonstrated in Table 9, isolating the backbone during this phase is vital for high-fidelity linguistic alignment. For both Chronos-2 and Toto, the *Frozen Backbone* configuration yields the peak LLM Eval Scores (67.5% and 63.7%, respectively), whereas full fine-tuning modes exhibit marked performance degradation. This indicates that freezing the encoder preserves the domain-specific sensor features learned during the refinement phase, mitigating catastrophic forgetting caused by noisy gradients propagating back from the LLM during caption generation.

Furthermore, employing a grounded output format—requiring the model to explicitly predict the activity label prior to generating the narrative—ensures the latent representation is anchored in the correct operational context. This grounding mechanism serves as a critical structural prior. As evidenced by the results, the high classification accuracy achieved by the frozen Chronos-2 (68.6%) and Toto (67.8%) architectures directly correlates with their superior caption generation scores. This confirms that explicitly grounding the temporal latents in an operational class is essential for generating reliable, physically consistent industrial reports.

*Table 9.* Stage-2: Multimodal TSFM Performance on DDR Captioning

| Model | TS Encoder Train Mode | Multi-stage MM-TSFM | Multi-stage TS Encoder Backbone | LLM Eval Score (%) | Activity Label Classification Accuracy (%) |
|---|---|---|---|---|---|
| OpenTSLM CNN (Default) | Full finetune | No | No | 59.8 | 58.7 |
| OpenTSLM CNN (Default) | Full finetune | Yes | No | **63.7** | 62.8 |
| OpenTSLM CNN (Default) | Full finetune | No | Yes | 62.0 | **62.9** |
| OpenTSLM CNN (Default) | Full finetune | Yes | Yes | 59.7 | 62.6 |
| OpenTSLM CNN (Default) | Frozen Backbone | Yes | Yes | 59.0 | 60.7 |
| OpenTSLM Chronos-2 | Full finetune | No | No | 61.2 | 61.5 |
| OpenTSLM Chronos-2 | Full finetune | Yes | No | 61.2 | 64.0 |
| OpenTSLM Chronos-2 | Full finetune | No | Yes | 62.5 | 64.4 |
| OpenTSLM Chronos-2 | Full finetune | Yes | Yes | 56.6 | 63.0 |
| OpenTSLM Chronos-2 | Frozen Backbone | Yes | Yes | **67.5** | **68.6** |
| OpenTSLM Toto | Full finetune | No | No | 61.1 | 60.1 |
| OpenTSLM Toto | Full finetune | Yes | No | 58.9 | 63.0 |
| OpenTSLM Toto | Full finetune | No | Yes | 60.3 | 58.9 |
| OpenTSLM Toto | Full finetune | Yes | Yes | 61.3 | 62.0 |
| OpenTSLM Toto | Frozen Backbone | Yes | Yes | **63.7** | **67.8** |

**I.3. Decoder Tuning and GCA Sufficiency**

To evaluate the optimal parameter-efficient fine-tuning (PEFT) strategy, we compared our isolated Gated Cross-Attention (GCA) approach against a configuration that additionally applied Low-Rank Adaptation (LoRA) (11) across the Gemma3-1B linguistic decoder. Incorporating LoRA resulted in a 2.3% performance degradation for the OpenTSLM Chronos-2 variant. This indicates that the interleaved GCA layers possess sufficient capacity to model the cross-modal coupling between the textual prompts and the continuous sensor latents. Introducing additional trainable parameters to the decoder likely induced over-fitting on the constrained lexical distribution of the DDR reports, thereby compromising the underlying LLM's generalized reasoning capabilities.

## I.4. Perceiver Resampler Configuration

The Perceiver Resampler serves as a crucial dimensionality bottleneck, mapping variable-length, high-frequency sensor sequences to a fixed-length set of temporal tokens. As demonstrated in Table 10, configuring the resampler to output 128 latent patches provides the optimal balance between data compression and feature fidelity. Aggressively restricting the bottleneck to 64 tokens results in measurable information loss (63.4% LLM Eval Score), failing to capture the full operational context. Conversely, expanding the patch count to 256 or higher introduces redundant temporal noise and unnecessarily consumes the cross-attention computational budget, which ultimately degrades the decoder's generation performance.

*Table 10.* Ablation on Perceiver Resampler Latent Patch Density

| Model | Perceiver Latent Patches | LLM Eval Score (%) |
|---|---|---|
| OpenTSLM Chronos-2 | 64 | 63.4 |
| OpenTSLM Chronos-2 | 128 | **67.5** |
| OpenTSLM Chronos-2 | 256 | 64.7 |
| OpenTSLM Chronos-2 | 512 | 65.1 |

## I.5. Output Grounding Validation

As hypothesized during the evaluation of our multi-stage training strategy (Section I.2), the structural formatting of the target sequence plays a significant role in generation fidelity. An isolated ablation of the grounded output format—where the model is explicitly forced to predict the activity classification label prior to generating the narrative text ($\langle$*Class label*$\rangle$ *DDR:* $\langle DDR \rangle$)—yielded a direct 3.3% performance improvement compared to ungrounded generation. This empirically confirms that explicitly anchoring the cross-modal latents with the operational context acts as a necessary structural prior before narrative generation begins.

## I.6. Prompt Specificity and Context Dynamics

During the DDR generation task, empirical results indicated an inverse relationship between prompt verbosity and narrative accuracy. Utilizing highly descriptive, over-specified prompts reduced the OpenTSLM Chronos-2 evaluation score from 67.5% to 65.8%. This degradation suggests that excessive instruction causes the model to prioritize rigid template adherence over the nuanced interpretation of the underlying sensor latents.

This behavior aligns with observations in sub-7B parameter language models, such as our utilized Gemma3-1B decoder. While massive LLMs can effectively navigate dense textual contexts, compact models are highly susceptible to "prompt noise" (10). Verbose prompts can precipitate information overload, diverting the model's limited attention mechanisms away from the cross-attended temporal tokens. Consequently, employing concise, information-dense prompts yields superior performance, allowing the architecture to allocate its computational budget toward processing the telemetry data rather than parsing redundant linguistic instructions. A comparative analysis of these prompting strategies and their respective outputs is provided in the Appendix (Section F.1).

## J. VLM Few-Shot Ablation Study

We evaluate the impact of few-shot prompting strategies on the Vision-Language Model (VLM) performance using a subset of 200 samples from the test set. Table 11 compares zero-shot inference against 3-shot and 9-shot settings using single fixed, per-class reference examples.

The results indicate that while increasing the number of shots provides a slight upward trend in evaluation scores—moving from 53.2% in zero-shot to 54.8% in a 9-shot configuration—the gains are marginal. Moreover, higher shot counts increase token overhead and inference costs. Because zero-shot prompting provides a competitive performance-to-cost ratio and simplifies the prompt architecture, it was selected as the primary configuration for the main experiments.

*Table 11.* Ablation of few-shot configurations for the VLM approach, including inference cost.

| Few-Shot Count | Score (%) | Cost / 100 Calls ($) |
|---|---|---|
| Zero-shot (Baseline) | 53.2 | **2.677** |
| 3-shot | 54.2 | 3.085 |
| 9-shot | **54.8** | 3.395 |

## K. Agentic Reasoning and Prompting Ablation

We analyze how the Agentic approach responds to variations in inference-time reasoning budgets and the inclusion of expert-level diagnostic prompting. This ablation is conducted on a subset of 200 samples from the test set to evaluate the model's sensitivity across our evaluation criteria. We compare three configurations: (i) a baseline agent, (ii) a high-thinking variant with an increased reasoning budget, and (iii) an expert-prompt variant where detailed instructions on DDR writing are provided.

*Table 12.* Ablation of the Agentic approach. High corresponds to stronger inference-time reasoning. Expert prompt uses detailed DDR-writing instructions.

| Model Variant | LLM Eval Score (%) | | | | | | | |
|---|---|---|---|---|---|---|---|---|
| | Weighted Score | A | B | C | D | E | F | G | H |
| Baseline | 69.7 | 67 | 99 | 100 | 51 | 87 | 48 | 62 | 52 |
| High Thinking | **70.2** | 72 | 98 | 100 | 52 | 84 | 42 | 60 | 51 |
| Expert Prompt | 68.5 | 67 | 99 | 100 | 47 | 85 | 51 | 63 | 49 |

Increasing the reasoning budget yields only a marginal improvement, from 69.7% to 70.2%. The improvement is mainly visible in the primary operation criterion (A). However, this comes with small degradations in parameter extraction (E) and secondary details (F–H), which limits the overall gain. In contrast, adding detailed expert instructions reduces performance to 68.5%. While this configuration slightly improves some specific attributes (e.g., hole size and BHA type), it degrades performance on operation coverage (D) and parameter matching (E).

These results suggest that the Agentic approach is not strongly limited by reasoning depth or prompt specificity. Instead, performance is primarily driven by the model's ability to retrieve and verify relevant information through tools.

## L. Agentic Tool-Calling Analysis

The DDR generation agent is equipped with a 14-tool API providing structured, read-only access to the multivariate sensor data of each time window. Across 4,622 evaluation cases, the agent issued 28,775 total tool calls (mean $6.2 \pm 2.3$, range 0–21 calls per case).

### L.1. Tool Inventory and Drilling-Domain Interpretation

Table 13 catalogs each tool alongside its observed usage frequency and its functional role in the drilling domain.

*Table 13.* Agentic tool inventory. *Cases (%)* is the fraction of the 4,622 evaluation cases in which the tool is invoked at least once. *Calls* is the total invocation count across all cases.

| TOOL | CATEGORY | CASES (%) | CALLS | DRILLING-DOMAIN ROLE |
|---|---|---|---|---|
| DESCRIBE_CONTEXT | ORIENTATION | 100.0 | 4,621 | READS THE TIME WINDOW, SAMPLING RATE, AND ACTIVE SENSOR LIST — THE AGENT'S FIRST SITUATIONAL-AWARENESS STEP |
| DESCRIBE_DATAFRAME_VIEW | ORIENTATION | 99.9 | 4,618 | INSPECTS THE DATA SCHEMA AND COLUMN TYPES TO CONFIRM DATA INTEGRITY BEFORE QUERYING |
| QUERY_DATAFRAME_VIEW | AGGREGATION | 72.2 | 9,371 | EXECUTES SQL AGGREGATIONS WITH ARBITRARY FILTERS; USED FOR COMPUTING AVERAGES OVER PUMP-ON PERIODS, DETECTING STEP CHANGES, OR ISOLATING ACTIVE DRILLING SUBINTERVALS |
| GET_SENSOR_STATISTICS | AGGREGATION | 66.8 | 3,790 | RETURNS MIN/MAX/MEAN/STD/PERCENTILES FOR A USER-SPECIFIED SENSOR GROUP OVER A TIME RANGE; EXTRACTS THE STANDARD DRILLING-PARAMETERS BLOCK (FLOW RATE, STAND PRESSURE, RPM, WOB, TORQUE, HOOKLOAD) |
| GET_DEPTH_SUMMARY | SPATIAL | 61.3 | 2,831 | REPORTS BIT DEPTH TRAJECTORY (START, END, MIN, MAX) AND HOLE DEPTH, QUANTIFYING FOOTAGE DRILLED OR PULLED AND THE DRILL-STRING STAND-OFF IN OPEN HOLE |
| GET_SENSOR_TREND | TEMPORAL | 18.5 | 1,133 | FITS PIECEWISE LINEAR SEGMENTS TO A SINGLE SENSOR; PRIMARILY APPLIED TO BIT DEPTH FOR COARSE ROP ESTIMATION AND TO TANK VOLUME FOR PIT-GAIN/LOSS MONITORING |
| GET_SENSOR_VALUES | RETRIEVAL | 10.8 | 846 | RETURNS A RESAMPLED RAW TIME SERIES FOR A FOCUSED SUBINTERVAL; USED FOR DETAILED CHARACTERIZATION OF TRANSIENT EVENTS SUCH AS BREAK-CIRCULATION OR WASHOUT |
| GET_VALUE_AT_TIME | RETRIEVAL | 9.5 | 903 | READS A POINT-IN-TIME SNAPSHOT OF SELECTED SENSORS; USED TO ANCHOR DDR ENTRIES AT THE START, END, OR KEY TRANSITION OF AN OPERATION |
| DETECT_ANOMALIES | ANOMALY | 9.1 | 421 | APPLIES A Z-SCORE THRESHOLD TO FLAG SUSTAINED SPIKE EVENTS; TARGETED AT HOOKLOAD (99.3% OF CALLS) AND TORQUE (85.3%) FOR OVERPULL, DRAG, AND STICK-SLIP SCREENING |
| COMPUTE_DERIVED_SIGNAL | DERIVED | 3.1 | 146 | COMPUTES DELTA (INCREMENTAL FOOTAGE $\approx$ ROP) OR GRADIENT (RATE-OF-CHANGE FOR TRANSITION DETECTION) FROM RAW SENSOR CHANNELS |
| COMPARE_SENSORS | CORRELATION | 1.4 | 63 | CROSS-CORRELATES TWO SENSORS AND REPORTS LAG AND SIMULTANEOUS ANOMALY CO-OCCURRENCE |
| RANK_TIME_WINDOWS | RANKING | 0.3 | 18 | SCORES FIXED-WIDTH SUB-WINDOWS BY ACTIVITY LEVEL; USED TO ISOLATE THE MOST OPERATIONALLY SIGNIFICANT SUBINTERVAL |
| GENERATE_SENSOR_PLOT | VISUALIZATION | 0.2 | 11 | RENDERS A MULTI-SENSOR TIME-SERIES PLOT FOR VISUAL INSPECTION OF AMBIGUOUS PATTERNS |
| COMPARE_INTERVALS | COMPARISON | 0.1 | 3 | CONTRASTS TWO TEMPORAL SUB-INTERVALS WITH PAIRED STATISTICAL SUMMARIES |

### L.2. Four-Phase Reasoning Workflow

Inspection of tool-transition sequences reveals a consistent four-phase structure across the dataset.

**Phase 1 — Temporal and Sensor Orientation (100 % of cases).** Every trace opens with describe_context followed immediately by describe_dataframe_view. The describe_context→describe_dataframe_view transition is the single most consistent adjacent pair (4,618 occurrences). This mirrors a drilling engineer's first action: confirming which surface channels are active and at what recording frequency before reading any value.

**Phase 2 — Spatial Grounding (61.3 % of cases).** get_depth_summary anchors the analysis in wellbore geometry, establishing start/end bit depth, total footage drilled or pulled, and the relationship between bit depth and hole depth (open-hole stand-off). Operations involving significant pipe movement (rotary drilling, tripping, running casing) invoke this phase more consistently than static operations (monitoring, cementing).

**Phase 3 — Statistical Profiling (≈99 % of cases).** The agent characterizes the full interval through iterative SQL aggregations (`query_dataframe_view`: 72.2 %) and multi-sensor statistical summaries (`get_sensor_statistics`: 66.8 %). The most frequent transition in the entire corpus is `query_dataframe_view`→`query_dataframe_view` (5,074 occurrences), reflecting progressive refinement: narrowing the time filter, adding `WHERE` clauses for pump-on periods, or computing new derived columns to isolate the active sub-interval from standstill. Typical queries extract the standard drilling-parameters block reported in every DDR: flow rate, stand pressure, rotation velocity, WOB, torque, and hookload.

**Phase 4 — Deep Investigation (33.4 % of cases).** Triggered when Phases 1–3 surface ambiguity or anomalous signals, the agent selectively invokes higher-resolution tools: (i) `get_sensor_trend` to estimate ROP from piecewise bit-depth slopes or to detect pit gain/loss from tank-volume trends; (ii) `detect_anomalies` with calibrated z-score thresholds for hookload overpull/drag and torque stick-slip screening; (iii) `compute_derived_signal` to reconstruct a coarse ROP signal as a delta or gradient of bit depth; (iv) `get_value_at_time` to anchor readings at the precise start, end, or transition point of a sub-operation.

### L.3. Illustrative Trace: Pull-Out-of-Hole with Hookload Screening

Table 14 presents the annotated 12-step trace for a 180-minute Pull-Out-of-Hole (POOH) operation (code `TRIP_DRILLC`; bit depth 5,984 ft → 4,020 ft). The agent correctly identifies the POOH trajectory, isolates a break-circulation event at $t \approx 119$ min, and screens for hookload anomalies, ultimately generating: *"POOH W/ DRILL STRING F/ 5,984' T/ 4,020'. BROKE CIRC 287 GPM/277 PSI."* (ground truth: *"POOH 5 1/2 DP WITH 12 1/4 RSS BHA ON ELEVATOR FROM 5984 FT TO 4020 FT"*).

*Table 14.* Annotated tool trace for a 180-minute POOH operation (`TRIP_DRILLC`; bit depth 5,984 ft → 4,020 ft). Each step is mapped to its drilling-context purpose.

| STEP | TOOL | DRILLING-CONTEXT PURPOSE |
|---|---|---|
| 1 | DESCRIBE_CONTEXT | CONFIRMS 180-MIN WINDOW (15:00–18:00 UTC), ≈1 HZ SAMPLING, 11 ACTIVE SURFACE SENSORS |
| 2 | DESCRIBE_DATAFRAME_VIEW | VERIFIES COLUMN SCHEMA; ALL STANDARD SURFACE SENSORS PRESENT |
| 3 | GET_DEPTH_SUMMARY | BIT DEPTH 5,984 FT → 4,020 FT; HOLE DEPTH ≈6,190 FT; CONFIRMS 1,964 FT POOH |
| 4 | GET_SENSOR_STATISTICS | PROFILES FLOW RATE (MAX 287 GPM), HOOKLOAD, TORQUE, STAND PRESSURE OVER THE FULL INTERVAL |
| 5 | GET_VALUE_AT_TIME ($t$=0) | START ANCHOR: TANK VOL 702.7 BBL, PUMPS OFF, STATIC STAND PRESSURE (25 PSI) |
| 6 | GET_VALUE_AT_TIME ($t$=180) | END ANCHOR: TANK VOL 674.7 BBL; 28 BBL DROP OVER 3 HR — NO ANOMALOUS PIT GAIN OR LOSS |
| 7 | COMPARE_SENSORS (BIT_DEPTH, HOOK_LOAD) | HOOKLOAD–DEPTH CORRELATION ($r=-0.10$); SURFACES CO-OCCURRING ANOMALY TIMESTAMPS FOR FURTHER INVESTIGATION |
| 8 | QUERY_DATAFRAME_VIEW | WHERE FLOW_RATE > 200: ISOLATES PUMPS-ON PERIOD AT $t \approx 119$–122 MIN (BREAK-CIRCULATION EVENT) |
| 9 | GET_SENSOR_TREND (BIT_DEPTH, 5 SEGMENTS) | PIECEWISE ROP: −542 FT/HR (STEADY POOH); CONFIRMS NO DRILLING-DOWN INTERVALS WITHIN THE WINDOW |
| 10 | GET_SENSOR_VALUES (FLOW_RATE, STAND_PRESSURE, $t$=116–122) | RETRIEVES DETAILED TIME SERIES FOR THE BREAK-CIRCULATION EVENT: 287 GPM AT 277 PSI |
| 11 | DETECT_ANOMALIES (HOOK_LOAD, $z$=2.5) | Z-SCORE SCREENING FOR OVERPULL/DRAG EVENTS DURING POOH; RETURNS SPIKE EVENTS FLAGGED ABOVE THRESHOLD |
| 12 | GET_VALUE_AT_TIME ($t$=119.2) | ANCHOR AT CIRCULATION START: BIT DEPTH 5,010 FT, HOOKLOAD 21.7 KLB, FLOW RATE 251 GPM |

### L.4. Sensor Targeting Patterns

**Anomaly detection.** Every `detect_anomalies` call targets hookload (99.3 % of calls) and in 85.3 % of cases co-selects torque, with stand pressure and flow rate as frequent corroborating channels. This reflects the domain standard: hookload deviations signal overpull and drag, torque oscillations signal stick-slip, and pressure/flow confirm whether the anomaly coincides with active circulation.

**Trend analysis.** `get_sensor_trend` is applied predominantly to bit depth (62.1 % of trend calls) for coarse ROP estimation and direction confirmation, and to tank volume (15.2 %) for passive pit-gain/loss monitoring. Remaining trend calls target flow rate (6.0 %), rotation velocity (5.3 %), and stand pressure (3.4 %).

**Derived signals.** Among `compute_derived_signal` invocations, the `delta` operation on bit depth (55.5 % of derived calls) reconstructs incremental footage per time step (a coarse ROP proxy), while `gradient` operations (40.4 %) detect abrupt transitions in pipe movement direction or pump rate.

## M. Extended Qualitative Examples

**Analysis of Extended Examples**

The examples detailed in Tables 15–16 illustrate the distinct operational profiles of the evaluated architectures. As shown in Table 15 (Strong Overall Performance, Refs 1 & 2), when all models succeed, they capture different facets of the same event: the Agentic framework provides exact numerical telemetry (e.g., 669 GPM/1,582 PSI), the TSFM offers an accurate operation label (e.g., motor drilling), and the VLM summarizes visual trends. Conversely, universal failures typically occur on stationary or subtle events. In Table 15 (Universal Failures, Ref 2), a simple static flow check causes all models to confidently hallucinate dynamic tripping or RIH activities.

Table 16 isolate the specific structural advantages of each model. For instance, (Agentic Strengths, Refs 1 & 3), the system accurately recovers the exact wash-down depths (6,865' to 7,065') and drill-out cement start depths (2,911') where the TSFM mislabels the activities entirely.

The TSFM, however, excels at recognizing temporally extended contextual events. Grounded in time-series morphology, it successfully identifies specific equipment semantics. In Table 16 (TSFM Strengths, Refs 1 & 4), the TSFM correctly recognizes a 13 3/8" casing run and a 7" liner assembly, whereas the Agentic model defaults to generic rotary or drill-string movements and misses the contextual purpose of the run.

Finally, while the VLM struggles with exact numerical grounding, it demonstrates a unique capability to translate visual step-changes into procedural narratives. As seen in Table 16 (VLM Strengths, Ref 2), it accurately logs sequential physical actions like pump starts and flow increases from visual plot patterns alone, explicitly confirming no losses occurred while the Agentic model hallucinated severe losses.

*Table 15.* Extended qualitative examples showing consistent successes, universal failures, and mixed outcomes.

| Category | Reference | VLM | Agentic | TSFM |
|---|---|---|---|---|
| **Strong Overall Performance** *All models scored well* | DRLG 12¼" W/ MOTOR BHA F/ 2262' TO 2343'. | **(77.0%):** DRILL AHEAD; ESTAB CIRC. FLOW ~80 GPM; RPM ~60; SPP ~1500 PSI; TQ 5–15 KFT-LB; WOB ~60 KLB. BIT DEPTH/MD INCREASING STEADILY; HOOKLOAD STABLE. NO ANOMALIES. | **(92.0%):** DRILL HOLE W/ ROTARY BHA F/ 2,262' T/ 2,343'. 669 GPM/1,582 PSI, 59 RPM, WOB 17 KLB. NO LOSSES. | **(92.0%):** DRILL 16" HOLE TO 2309'. |
| | CIRC CLEAN HOLE. Observed seepage losses. | **(70.0%):** CIRC @ TD ~50 MIN; NO ROTATION. RPM 0, TQ 0. FLOW ON THEN SHUT DOWN; SPP DECLINING DURING CIRC. WOB REDUCING; HL INCREASING. PIT GAIN AFTER PUMP-OFF. P/U AND START POOH; BIT LIFTED OFF BTM. | **(85.0%):** CIRCULATED HOLE CLEAN OFF BTM. BIT F/4566' T/4551'. 210-231 GPM / 27-92 PSI. NO ROTATION. | **(85.0%):** CIRCULATE LINER VOLUME. |
| **Universal Failures** *All models scored poorly* | DRILLOUT HARD CEMENT ON SHOE TRACK F/ 9193 FT T/ 9366 FT | **(25.0%):** DRILLING AHEAD; BIT DEPTH INCREASING. FLOW ~200 GPM; SPP ~1200 PSI; RPM ~60; WOB TO ~16 KLB; TQ SPIKES TO ~30 KFT-LB. BRIEF PUMP/ROTATION STOPS FOR CONNECTIONS. PITS STABLE, NO LOSSES. HOOKLOAD STEADY; NO OVERPULL. | **(40.0%):** REAM DOWN W/ ROTARY BHA F/ 9,154' T/ 9,365'. DRILL HOLE F/ 9,365' T/ 9,368' - 206 GPM/1400 PSI, 50 RPM, WOB 11 KLB, TQ 4.8 KFTLB. | **(23.0%):** WORK ON 10 FT. OBSERVED SLIPS WITH LINER WHILE WASHING DOWN, WASHING DOWN, LAST STAND PRIOR TO WASHING DOWN W/ RSS BHA. |
| | RAN 9 5/8" CASING TO 1115FT. | **(35.0%):** RIH BHA F/0' T/~900'. Intermittent fill/circ; flow peaks ~120 gpm. No rotation; torque ~0; HHP ~0. Bit depth stepwise; block cycling. SPP low/variable. No clear anomalies (no overpull/slack-off spikes or losses evident). | **(35.0%):** CONT RIH W/ BHA F/ 40' T/ 939'. FILLED STRG. | **(20.0%):** CONT'D P/U, M/U 17½" LNR DPs 1x16in. WASH DOWN W/ LINER ASSY. WHILE FILLING EVERY 15 STANDS. |
| **Mixed Performance** *Inconsistent outcomes across models* | DRILL 6" HOLE W/ BHA F/17700' T/18296' | **(65.0%):** ROTARY DRILL AHEAD; REGULAR CONNECTIONS. BIT/WELL DEPTH INCREASING; BLOCK POSITION CYCLIC. PUMPS ON DURING DRILLING; RPM HIGH; WOB APPLIED; TORQUE AND SPP STABLE WITH CYCLIC DROPS ON CONNECTIONS. HOOKLOAD CYCLIC WITHOUT OVERPULL/SLACK-OFF EVENTS. FLOW OUT TRACKS FLOW IN; NO CLEAR LOSSES/GAINS OBSERVED. | **(95.0%):** DRILL HOLE F/ 17,700' T/ 18,296'. 231 GPM/2072 PSI, 99 RPM, WOB 10 KLB, 13 KFTLB. | **(95.0%):** DRILL 8 1/2" WITH RSS BHA TO 18383 FT. WITH CONTROLLED PARAMETERS |
| | SLIDE DRILL 26" HOLE W/ MOTOR BHA TO 1074'. WOB: 5-15 KLBS — GPM: 933 | **(62.0%):** DRILL AHEAD W/ CIRC. Bit depth increasing steadily; hole depth tracking; block lowering. SPP 1140–1320 psi w/ brief mid-interval dip then recovery. Flow-out 0.28–0.34 frac. Hookload variable trending down; WOB gradually increasing. RPM/TQ low most of interval then multiple spikes toward end. Mud tank volume slow fall then sharp step-up near 2 hr. No evident overpull/slack-off. | **(80.0%):** DRILL HOLE W/ MOTOR/MWD BHA F/ 1,031' T/ 1,075'. 793 GPM/1,231 PSI. 0 RPM. | **(77.0%):** DRILLED 17.5" HOLE 1054' TO 1060'. |

*Table 16.* Extended qualitative examples: Agentic, TSFM, and VLM strengths in various operational contexts. **Bold scores** mark the best-performing model per example; **bold text** highlights phrases in the winning prediction that directly correspond to information present in the reference entry. *Italics* draw attention to the contrasting misidentification in other models.

| Category | Reference | VLM | Agentic | TSFM |
|---|---|---|---|---|
| **Agentic Strengths** *Precise depth extraction and exact operation grounding via tool-verified retrieval* | CONT. DRILL CEMENT TO 9505 FT. D/O F.S F/9505' TO 9507'. D/O CEMENT IN RAT HOLE F/9507' TO 9510'. DRILL 10 FT INTO FORMATION TO 9520'. | (85.0%): *DRILL* F/ ~9440' T/ ~9560'. FR ~300 gpm — RPM 50–80 — WOB 6–12 klbf — SPP ~1500 psi. Brief pump/rotation shutdowns for connections. | **(85.0%):** DRILL HOLE W/ ROTARY BHA F/ 9,453' T/ **9,520'.** 224 GPM/1393 PSI, 57 RPM, WOB 8 KLB, 3.8 KFTLB. | (55.0%): *DRILL 8.5in SEC WITH RSS BHA TO 9498'.* |
| | DRILLED 12.25in HOLE W/ RSS BHA F/9588' T/9600'. | (62.0%): *DRILLING AHEAD.* FLOW ~600 GPM; RPM ~100; SPP 2700–2800 PSI; WOB 38–40 KLB. DEPTH INCREASING STEPWISE. | **(95.0%):** DOWN LNK, **RSS DRILLED HOLE F/ 9,588' T/ 9,600'.** 585 GPM/2744 PSI, 109 RPM, WOB 29 KLB, TORQ 10 KFTLB. | (32.0%): *DRILL OUT SHOE TRACK F/9567 FT T/9611 FT.* DRILL OUT RAT HOLE TO 9602 FT. |
| | POOH W/ 6" RSS-LWD-MWD BHA WHILE REAM-UPWARDS FROM 12083 FT TO 11901 FT. | (77.0%): POOH; BACKREAM/WASH EARLY AND LATE. WHEN PUMPING: ~240 GPM, SPP ~2,400 PSI, 70–80 RPM. TORQUE SPIKES TO ~32 KFT-LB. | **(92.0%): BROOH F/ 12,082' T/ 11,901'.** 250 GPM/2510 PSI, 71 RPM, 5 KFTLB. | (62.0%): BROOH REAMING DOWN F/ 11943 FT TO 11999 FT. |
| | POOH 12¼" MOTOR BHA ON 5-1/2" DPS F/ 6186' T/ 4400'. | (0.0%): *POOH BHA F/~5,700' T/~4,700'. Intermittent wash/ream while tripping:* flow ~500 gpm, SPP ~800–950 psi. | **(77.0%):** POOH W/ DRILL STRING F/ **5,998' T/ 4,492'.** BACKREAMED F/ 4,975' T/ 4,598' @ 509 GPM/723 PSI, 45 RPM. | (32.0%): *BACK REAMING FROM 4009 FT TO 4712 FT.* |
| **TSFM Strengths** *Contextual event recognition via implicit temporal morphology: casing, flow checks, and cementing* | RIH WITH 13-3/8" CSG TO 2830'. | (20.0%): *CONT DRILL* F/ ~500' TO ~1,500'. Flow ~200 gpm; RPM ≤16; TQ ≤16 kft-lb; WOB 0–140 klbf; HL 40–125 klbf. | (35.0%): CONT *RIH W/ DRILL STRING + BHA* F/ 839' TO 1,544'. FILLED STRING; PUMPED UP TO 208 GPM. | **(80.0%): RIH** 9-5/8" **CASING** FROM 982 FT TO 1630 FT. |
| | CONT'D RAN 9-5/8" CASING TO 6083 FT. | (20.0%): *RIH STRING* F/ 5,200' T/ 6,000'+. HOLE DEPTH STATIC. INTERMITTENT CIRC; SPP ~150–450 PSI. TORQUE ~0; WOB ~0. | (35.0%): CONT *RIH W/ DRILL STRING BHA* F/ 5,222' T/ 6,048'. NO ROTATION; PUMPED TO FILL. | **(100.0%): CONT. RIH WITH CRT 9-5/8" CSG** L-80 CRT F/5410 FT **TO 6135 FT** (36/60 JOINTS) |
| | MADE FLOW CHECK. | (40.0%): *WASH & REAM DOWN* F/ 10,020' TO 10,380' @ ~120 RPM; FLOW ON. STOP PUMPS; STATIC; BRIEF FLOW-OUT SPIKE. *POOH W/O CIRC* F/ 10,380' TO 10,020'. | (55.0%): *REAM DOWN W/ MOTOR/MWD BHA* F/ 10,299' T/ 10,376'. 496 GPM/2808 PSI, 120 RPM. | **(100.0%): FLOW CHECK — WELL IS STATIC** |
| | Performed 9-5/8" casing top job. | (35.0%): *WELL STATIC. NO CIRC* (0 GPM) — 0 RPM — 0 TORQUE — 0 SPP. DEPTHS HELD CONSTANT. TANK VOL STABLE. | (35.0%): *FLOW CHECK. WELL STATIC. NO LOSSES.* | **(95.0%): PERFORM TOP CMT JOB** |
| **VLM Strengths** *Procedural summaries from visual step-changes and trend patterns* | WASH DOWN F/ 9356 FT TO 9461 FT AT 207 GPM, 1300 PSI | **(85.0%):** RIH & **WASH DOWN**; HOLE DEPTH STATIC. BIT DEPTH F/ ~**9,385' T/** ~**9,440'. PUMP** ~**200 GPM; SPP** ~**1,200–1,300 PSI.** RPM 20–25 INTERMITTENT; WOB 0. TORQUE SPIKE ~25 KFT-LB. | (55.0%): WASH/REAM DOWN F/ 9,353' TO 9,442'. 161 GPM/816 PSI, 30 RPM, 0 WOB. *OBSERVED PARTIAL LOSSES.* | (55.0%): *M/UP HWDPS & CIRC. WITH RECIPROCATE STRING* |
| | CONDUCTED SHALLOW TEST — OK [210 GPM – 1850 PSI]. | **(70.0%):** CIRCULATE OFF-BTM; **STEP-RATE PUMP TEST. FLOW RAMPED IN STAGES; SPP ROSE TO STEADY PLATEAU.** RPM MOSTLY 0; HOOKLOAD HIGH/STABLE, WOB ~0. ACTIVE PIT VOL DROPPED THEN RECOVERED. SHUT PUMPS; P/U BLOCK. | (25.0%): *CIRCULATED HOLE CLEAN.* 152 GPM/1098 PSI. | (25.0%): *PERFORMED 7" LINER CMT JOB* |

