# OpenReview forum: "From Time-Series to Text: Multimodal and Agentic Approaches for Automated Drilling Report Generation"
_ICML.cc/2026/Workshop/FMSD — FMSD @ ICML 2026 Poster_

### Official Review · Reviewer_pKbk · 2026-05-18
**Interesting Three-Way Comparison, but the TSFM-Side Conclusions Are Not Yet Fully Supported**

**Rating:** 5
**Confidence:** 3

**Review:**

**Summary**

This paper studies automated Daily Drilling Report generation from raw multivariate sensor telemetry by comparing three paradigms: a TSFM-based multimodal model, a VLM operating on plotted signals, and a tool-augmented agentic system. The main empirical finding is that the agentic approach performs best overall, especially on numerically grounded aspects of the task, while the TSFM-based approach appears to offer some advantages on operation-level contextual recognition at much lower inference cost.

**Strengths**

1. The paper addresses a realistic and practically important structured-data problem with clear industrial relevance.
2. I found the three-way comparison between a TSFM-based multimodal model, a VLM, and a tool-augmented agentic system informative.
3. The strongest empirical result, namely the agentic approach’s advantage on numerically grounded report generation, is clearly supported by the reported rubric scores.
4. The paper also reports inference cost and latency, which is valuable and well aligned with the workshop’s priorities.

**Areas for Improvement**

1. The paper’s central interpretive term `temporally extended contextual events` is not defined clearly enough, which makes the TSFM-side claims harder to evaluate precisely.
2. In Secs. 5.1–5.2, the paper links class-level gains (e.g., Cementing and Casing) to rubric-level strengths (e.g., BHA type and operational details), but this connection is not shown explicitly enough.
3. The paper’s discussion of hybrid systems would benefit from slightly more careful framing. The abstract and conclusion sugget that hybrid systems are the clear next step, whereas the current results show complementary strengths between the agentic and TSFM-based approaches rather than directly demonstring that a hybrid system would be superior.

**Detailed Comments**

1. For the TSFM-side interpretation, I would have appreciated a clearer mapping between the term `temporally extended contextual events` and the actual evaluation evidence. For example, it would help to state more explicitly which rubric categories, operational classes, or qualitative examples are intended to support this notion, and what the underlying reasoning is.
2. In Secs. 5.1–5.2, the paper argues that the TSFM’s gains in classes such as Cementing and Casing stem from rubric-level strengths such as BHA type and operational details. This interpretation is plausible, but I would have found it more convincing if the paper more explicitly showed how those rubric-level improvements translate into the class-level gains.

**Justification of Score**

The paper addresses a relevant and practically meaningful structured-data problem, and I found the comparison between a TSFM-based multimodal model, a VLM, and an agentic system useful. The strongest result, namely the agentic approach’s advantage on numerically grounded report generation, is supported.

My main concern is that the TSFM-side conclusions are less tightly supported. In particular, `temporally extended contextual events` is not defined clearly enough, and the connection between rubric-level gains and operational-class improvements remains somewhat implicit. The hybrid-system conclusion also reads more as a plausible future direction than as a directly demonstrated finding.

---

### Official Review · Reviewer_8SzD · 2026-05-22
**Review of paper 69**

**Rating:** 7
**Confidence:** 4

**Review:**

## Summary
This paper systematically evaluates three paradigms for generating structured Daily Drilling Reports (DDRs) from raw, multivariate surface and downhole sensor telemetry: (i) a multimodal Time-Series Foundation Model (TSFM) architecture (based on OpenTSLM-Flamingo with a frozen Chronos-2 or Toto backbone and a frozen Gemma decoder with interleaved gated cross-attention layers), (ii) a Vision-Language Model (VLM) reasoning over 2D plots of telemetry data, and (iii) a tool-augmented ReAct-style agent querying raw sensor data via SQL (DuckDB) and analytical tools. The study finds a functional division of labor: the agent excels at exact numerical grounding (achieving 70.1% overall and near-perfect depth matching) but is computationally expensive, while the TSFM model excels at detecting complex operational states, context, and BHA metadata (achieving 67.5%) at a fraction of the cost, pointing to hybrid systems as a key future direction.

## Pros
- **Gated Cross-Attention & Two-Stage Training:** A clever, gradual update strategy that successfully bridges the time-series and language modalities. Freezing the refined TSFM backbone in Stage 2 effectively prevents noisy text-generation gradients from corrupting the specialized temporal features learned during pre-training.
- **Relational SQL Tool-Augmentation with Strict Verification:** Integrating a DuckDB SQL querying engine is a highly robust solution for reasoning over massive temporal sequences. Pairing this with a strict anti-hallucination constraint (requiring external tool verification for all reported numbers) successfully guarantees excellent numerical fidelity.

## Cons
- **No Empirical Validation of a Hybrid System:** Despite concluding that a hybrid system is the future, the authors do not implement one. The study would be much stronger if they evaluated a hybrid agent that uses a pre-trained TSFM *as a tool* within the ReAct loop to guide SQL queries rather than keeping both paradigms strictly segregated.
- **Lossy VLM Baseline:** Rendering high-frequency 1 Hz telemetry (up to 86k timesteps) into a static 2D image is an extremely lossy representation that structurally guarantees numerical failure. Comparing this degraded visual baseline against an agent with direct SQL access to raw data represents a "strawman" evaluation.